# *Closo*-Carboranyl- and Metallacarboranyl [1,2,3]triazolyl-Decorated Lapatinib-Scaffold for Cancer Therapy Combining Tyrosine Kinase Inhibition and Boron Neutron Capture Therapy

**DOI:** 10.3390/cells9061408

**Published:** 2020-06-05

**Authors:** Marcos Couto, Catalina Alamón, María Fernanda García, Mariángeles Kovacs, Emiliano Trias, Susana Nievas, Emiliano Pozzi, Paula Curotto, Silvia Thorp, María Alejandra Dagrosa, Francesc Teixidor, Clara Viñas, Hugo Cerecetto

**Affiliations:** 1Grupo de Química Orgánica Medicinal, Instituto de Química Biológica, Facultad de Ciencias, Universidad de la República, Iguá 4225, 11400 Montevideo, Uruguay; cataalamon@gmail.com; 2Institut de Ciència de Materials de Barcelona (ICMAB-CSIC), Campus UAB, 08193 Bellaterra, Spain; teixidor@icmab.es; 3Área de Radiofarmacia, Centro de Investigaciones Nucleares, Facultad de Ciencias, Universidad de la República, Mataojo 2055, 11400 Montevideo, Uruguay; fgmelian@gmail.com; 4Laboratorio de Neurodegeneración, Institut Pasteur de Montevideo, Mataojo 2020, 11400 Montevideo, Uruguay; mkovacs@pasteur.edu.uy (M.K.); etrias@pasteur.edu.uy (E.T.); 5Department of Boron Neutron Capture Therapy, National Atomic Energy Commission (CNEA), Avenida General Paz 1499, 1650 San Martín, Argentina; susanaisabelnievas@gmail.com; 6Department of Research and Production Reactors, CNEA, Presbítero Juan González y Aragón, 15, B1802AYA Ezeiza, Argentina; epozzi@cnea.gov.ar (E.P.); curotto@cae.cnea.gov.ar (P.C.); 7Department of Instrumentation and Control, CNEA, Presbítero Juan González y Aragón, 15, B1802AYA Ezeiza, Argentina; thorp@cae.cnea.gov.ar; 8CNEA, Avenida General Paz 1499, 1650 San Martín, Buenos Aires, Argentina; Consejo Nacional de Investigaciones Científicas y Técnicas (CONICET), Godoy Cruz 2290, 1425 CABA, Argentina; mariaalejandradgrs3@gmail.com

**Keywords:** tyrosine kinase inhibitors, lapatinib, [1,2,3]triazolyl linker, boron clusters, in vitro BNCT effect

## Abstract

One of the driving forces of carcinogenesis in humans is the aberrant activation of receptors; consequently, one of the most promising mechanisms for cancer treatment is receptor inhibition by chemotherapy. Although a variety of cancers are initially susceptible to chemotherapy, they eventually develop multi-drug resistance. Anti-tumor agents overcoming resistance and acting through two or more ways offer greater therapeutic benefits over single-mechanism entities. In this study, we report on a new family of bifunctional compounds that, offering the possibility of dual action (drug + radiotherapy combinations), may result in significant clinical benefits. This new family of compounds combines two fragments: the drug fragment is a lapatinib group, which inhibits the tyrosine kinase receptor activity, and an icosahedral boron cluster used as agents for neutron capture therapy (BNCT). The developed compounds were evaluated in vitro against different tyrosine kinase receptors (TKRs)-expressing tumoral cells, and in vitro–BNCT experiments were performed for two of the most promising hybrids, **19** and **22**. We identified hybrid **19** with excellent selectivity to inhibit cell proliferation and ability to induce necrosis/apoptosis of glioblastoma U87 MG cell line. Furthermore, derivative **22**, bearing a water-solubility-enhancing moiety, showed moderate inhibition of cell proliferation in both U87 MG and colorectal HT-29 cell lines. Additionally, the HT-29 cells accumulated adequate levels of boron after hybrids **19** and **22** incubations rendering, and after neutron irradiation, higher BNCT-effects than **BPA**. The attractive profile of developed hybrids makes them interesting agents for combined therapy.

## 1. Introduction

Tyrosine kinase receptors (TKRs) are transmembrane-type receptors with cytoplasmic tyrosine kinase domains, which transduce extracellular signals to a variety of intracellular signaling cascades involved in proliferation and differentiation of both normal and malignant cells [1]. The overexpression of some TKRs and their enhanced signaling contribute to the initiation, progression, and invasiveness of cancers [2]. Thus, TKRs are attractive targets for the development of therapeutic tools, for example, small inhibitors such as erlotinib (**Erl**)-targeting epidermal growth factor receptor (EGFR, ErbB1), and lapatinib (**Lap**) targeting EGFR and human epidermal growth factor receptor 2 (ErbB2, HER2) (Figure 1) [3,4,5]. Another anti-cancer strategy, boron neutron capture therapy (BNCT), has been recognized as a promising therapy for melanoma, locally malignant gliomas, and head and neck cancers [6,7,8,9,10]. BNCT is based on the nuclear capture and fission reactions of the ^10^B atom with low energy thermal/epithermal neutrons to yield high linear energy transfer α particles and recoiling ^7^Li nuclei. Since the path lengths of the particles are approximately 9−10 μm, similar to the dimensions of a single cell, ^10^B-containing cells are selectively destroyed by BNCT. Two compounds, *p-*borono-L-phenylalanine (**BPA**) and disodium mercapto-*closo*-undecahydrododecaborate (**BSH**), are clinically used in treatment of cancer with BNCT (Figure 1) [11,12]. **BPA** is selectively taken up into tumor cells, while **BSH** tumor selectivity is slightly low [13]. However, **BSH** and its derivatives are of increasing interest as boron carriers for BNCT, due to the ability to deliver large amounts of ^10^B atoms to tumor cells (12 times more B per **BSH**-molecule than **BPA**) [14,15,16]. The first boron drug Steboronine^®^ to be utilized in BNCT was recently developed by Stella Pharma corporation on 25 March 2020 [17]. It not only provides another option to the oncologists but benefits locally unresectable recurrent or unresectable advanced head and neck cancer treatment.

We have previously designed hybrid agents by combining substructures derived from **Erl** and icosahedral boron clusters with the aim to develop a new bimodal therapy of cancer. These bifunctional compounds, which combine the TKR-interaction/inhibition ability plus the selective and high boron-atoms-loading capacity in cells for the BNCT process, would act as anticancer bimodal agents (chemo- + radiotherapy) to result in significant clinical benefits such as reducing the doses to get the same therapeutic effect while diminishing the side effects suffered by the patient. We have demonstrated that the incorporation of the boron cluster has resulted in hybrids with enhanced and selective in vitro and in vivo anti-tumoral activities, i.e., compounds **1**, **2**, and **4** (Figure 1) [18,19,20,21]. From a structural point of view, relevant aspects were observed; the [1,2,3]triazolylalkyl-linker yielded hybrids with the most promising profiles, that is, being better than other links (see bio-profiles of **3** and **5**, Figure 1). Herein, we describe the exploration of the EGFR and ErbB2 inhibitor **Lap** as scaffold to develop new hybrid molecules for bimodal therapy of cancer treatment. These newly synthesized hybrids were evaluated in vitro as cytotoxic agents on TKRs-overexpressing cells, and its selectivity against glioma-cells was also tested. The cellular death mechanism triggered by one of these hybrids was also studied on glioma cells and astrocytes. Additionally, for selected hybrids, the in vitro ability to inhibit active EGFR in a cell-free system was performed. Moreover, BNCT potentiality was evaluated analyzing cellular B-accumulations followed by neutron irradiation experiments.

## 2. Materials and Methods

### 2.1. Chemistry

Chemicals were reagent-grade and were used as received from commercial suppliers (Merck (Sigma-Aldrich), Darmstadt, Germany). 1,2-*closo*-C_2_B_10_H_12_, and B_10_H_14_ were obtained from Katchem (Prague, Czech Republic) and **Lap** from Baoji Guokang Bio-Technology Co (Baoji, China). 1-CH_3_-1,2-*closo*-C_2_B_10_H_11_ was synthesized from B_10_H_14_ as reported in the literature [22]; 1,7-*closo*-C_2_B_10_H_12_ and 1-CH_3_-1,7-*closo*-C_2_B_10_H_11_ were synthesized from 1,2-*closo*-C_2_B_10_H_12_ and 1-CH_3_-1,2-*closo*-C_2_B_10_H_11_, respectively, following the reported procedures [23]; and Cs[3,3’-Co(1,2-C_2_B_9_H_11_)_2_] (**14**) [24], [3,3’-Co(8-(CH_2_CH_2_O)_2_-1,2-C_2_B_9_H_10_)(1’,2’-C_2_B_9_H_11_)] (**15**) [25,26], and [3,3’-Co(8-N_3_-(CH_2_CH_2_O)_2_-1,2-C_2_B_9_H_10_)(1’,2’-C_2_B_9_H_11_)] (**16**) [27] were synthesized as reported,. Intermediates **6–13** were synthesized as reported in the literature [18,28]. Most reactions were performed under an atmosphere of nitrogen by employing standard Schlenk techniques. Analytical thin-layer chromatography (TLC) was carried out on pre-coated plates with silica gel 60 on aluminum foil F_254_ (Merck). Compounds were visualized by staining using UV light (254 nm) and/or by a 0.5% acidic solution of PdCl_2_ in HCl/methanol for boron-containing derivatives.

### 2.2. Instrumentation

Elemental analyses were performed using a Carlo Erba Model EA1108 elemental analyzer instrument (AB, Canada). All NMR spectra were recorded on Bruker ARX-300 or on Bruker DPX-400 spectrometers (Billerica, MA, USA) equipped with the appropriate decoupling accessories. The ^1^H (300.13 MHz or at 400.13 MHz), ^11^B{^1^H} (96.29 MHz), and ^13^C{^1^H} (75.47 MHz or at 100.77 MHz) NMR spectra (see Appendix A) were recorded in CDCl_3_, CD_3_COCD_3_, or (CD_3_)_2_SO at 298 K. Chemical shifts are reported in units of parts per million downfield from the reference, and all coupling constants are reported in Hertz (Hz). The ^11^B and ^11^B{^1^H} NMR shifts were referenced to external BF_3_·OEt_2_, while the ^1^H, ^1^H{^11^B}, and ^13^C{^1^H} NMR shifts were referenced to SiMe_4_. Multiplicity is abbreviated as follows: s is the singlet, d is the doublet, dd is the doublet of doublet, dt is doublet of triplets, dq is doublet of quartets, t is the triplet, m is the multiplet, and bs is the broad singlet. MS were performed at a Shimadzu QP-2010 spectrometer (Kyoto, Japan) at 70 eV ionizing voltage. Mass spectra are presented as *m*/*z* (% rel int.). MALDI-TOF mass spectra were recorded in the negative-ion mode using a Bruker Biflex MALDI-TOF (N_2_ laser; λ_exc_ = 337 nm; 0.5 ns pulses); voltage ion source 20.00 kV (Uis1) and 17.50 kV (Uis2)). UV measurements were performed on spectrofluorometer Varioskan flash, Thermo^®^ (Waltham, MA, USA) at 298 K and using 1.0 cm cuvettes.

### 2.3. Synthesis of Lapatinib Derivative

Triethylamine (1 equiv., 0.1 mL, 0.69 mmol) was added drop by drop to a stirred suspension of **Lap** (1 equiv., 400 mg, 0.69 mmol) in CHCl_3_ (12 mL). The mixture was stirred for 1 h at room temperature. After that, 3-bromo-1-propyne solution (80% in toluene, 1.05 equiv., 0.075 mL, 0.72 mmol) was added over a period of 15 min. The mixture was stirred overnight at reflux, and then it was quenched with an aqueous saturated solution of NH_4_Cl (15 mL) and extracted with CHCl_3_ (3 × 20 mL). The organic layer was dried over MgSO_4_ and evaporated in vacuum to dryness. The orange residue was purified by SiO_2_ column chromatography (CH_2_Cl_2_:MeOH, 97:3) to give the desired compound as a yellow solid (398 mg, 74%). ^1^H-NMR (400 MHz, CDCl_3_) δ: 8.69 (s, 1H, pyrimidine-H), 8.40 (bs, 2H, -NH and Ar-H), 7.95 (dd, *J*_H,H_ = 8.7; 1.5, 1H, Ar-H), 7.90 (d, *J*_H,H_ = 2.5, 1H, Ar-H), 7.85 (d, *J*_H,H_ = 8.7, 1H, Ar-H), 7.67 (dd, *J*_H,H_ = 8.8; 1.5, 1H, Ar-H), 7.40–7.33 (m, 1H, Ar-H), 7.26–7.20 (m, 2H, Ar-H), 7.04 (dd, *J*_H,H_ = 8.7; 2.1, 1H, Ar-H), 6.99 (d, *J*_H,H_ = 8.7, 1H, Ar-H), 6.71 (d, *J*_H,H_ = 3.3, 1H, furyl-H), 6.39 (d, *J*_H,H_ = 3.3, 1H, furyl-H), 5.16 (s, 2H, -ArO-CH_2_-Ar), 3.87 (s, 2H, -N-CH_2_-furyl), 3.46 (d, *J*_H,H_ = 2.3, 2H, -N-CH_2_-C≡CH), 3.37 (bs, 4H, -N-CH_2_-CH_2_-SO_2_), 2.97 (s, 3H, -SO_2_CH_3_), 2.33 (t, *J*_H,H_ = 2.2, 1H, -N-CH_2_-C≡C). ^13^C{^1^H}-NMR (100.77 MHz, CDCl_3_) δ: 164.7, 161.4, 158.0, 154.9, 153.1, 150.9, 149.5, 144.8, 139.3, 132.8, 130.2, 129.0, 128.7, 125.3, 123.1, 122.7, 122.5, 115.9, 115.0, 114.7, 114.3, 113.9, 111.8, 106.9, 74.5, 70.50, 55.1, 52.4, 49.4, 48.3, 46.8, 42.1.

### 2.4. General Procedure for Hybrids **18**–**23** Preparation

Lapatinib derivative (1 equiv.) dissolved in THF:H_2_O (1:1, *v*/*v*) (6 mL for each 50 mg of Lapatinib derivative) was introduced into a 50-mL round-bottom flask equipped with a magnetic stirring bar. Then, sodium ascorbate (15 mol %), CuSO_4_·5H_2_O (10 mol %) and the corresponding azide (**8**, **9**, **12**, **13**, **16,** or **17**) (1.1 equiv.) were added. The mixture was stirred overnight at room temperature. After that, the solvent was evaporated and the crude was dissolved in CH_2_Cl_2_ (15 mL for each 50 mg of lapatinib derivative) and washed with brine (3 × 5 mL for each 50 mg of lapatinib derivative). The organic layer was dried over MgSO_4_ and evaporated in vacuum to dryness. The product was purified by preparative SiO_2_-TLC or chromatographic column using CH_2_Cl_2_:MeOH (97:3) as eluent.

#### 2.4.1. Hybrid **18**

Yellow solid (123 mg, 75%). ^1^H{^11^B}-NMR (400 MHz, CDCl_3_) δ: 8.68 (s, 2H, pyrimidine-H and -NH), 8.51 (s, 1H, Ar-H), 7.92 (d, *J*_H,H_ = 8.8, 1H, Ar-H), 7.85 (m, 2H, Ar-H), 7.72 (dd, *J*_H,H_ = 8.7; 2.1, 1H, Ar-H), 7.54 (s, 1H, triazole-H), 7.52–7.42 (m, 1H, Ar-H), 7.24 (dd, *J*_H,H_ = 15.3; 8.0, 2H, Ar-H), 7.04 (dd, *J*_H,H_ = 8.5; 1.6, 1H, Ar-H), 6.98 (d, *J*_H,H_ = 8.9, 1H, Ar-H), 6.68 (d, *J*_H,H_ = 3.2, 1H, furyl-H), 6.36 (d, *J*_H,H_ = 3.2, 1H, furyl-H), 5.14 (s, 2H, -ArO-CH_2_-Ar), 4.32 (t, *J*_H,H_ = 6.2, 2H, triazole-CH_2_-CH_2_- CH_2_-C_cluster_), 3.88 (s, 2H, -N-CH_2_-furyl), 3.79 (s, 2H, -N-CH_2_-triazole), 3.61 (s, 1H, C_cluster_-H), 3.35 (t, *J*_H,H_ = 6.4, 2H, CH_2_-CH_2_-SO_2_), 3.22 (t, *J*_H,H_ = 6.5, 2H, -N-CH_2_-CH_2_-), 2.97 (s, 3H, -SO_2_CH_3_), 2.28–2.15 (bs, 4H, triazole-CH_2_-CH_2_-CH_2_-C_cluster_). ^13^C{^1^H}-NMR (100.77 MHz, CDCl_3_) δ: 164.6, 161.4, 158.0, 154.8, 153.0, 150.9, 149.3, 144.9, 139.2, 132.7, 130.2, 130.1, 128.9, 128.8, 128.6, 125.3, 122.9, 122.5, 115.8, 115.6, 115.0, 114.8, 114.2, 113.8, 111.8, 107.0, 73.6, 70.5, 61.8, 52.3, 49.2, 49.1, 48.66, 46.5, 42.3, 34.8, 29.7. ^11^B{^1^H}-NMR (96.29 MHz, CDCl_3_) δ: −2.5 (1B), −5.9 (1B), −9.5 (2B), −12.2 (6B). Anal. calcd. for: C_37_H_45_B_10_ClFN_7_O_4_S: C, 52.50; H, 5.36; N, 11.58. Found: C, 52.39; H, 5.68, N, 11.95.

#### 2.4.2. Hybrid **19**

Yellow solid (191 mg, 84%). ^1^H{^11^B}-NMR (400 MHz, CDCl_3_) δ: 8.68 (s, 2H, pyrimidine-H and -NH), 8.52 (s, 1H, Ar-H), 7.93 (dd, *J*_H,H_= 8.8; 1.1, 1H, Ar-H), 7.90 (d, *J*_H,H_ = 2.5, 1H, Ar-H), 7.84 (d, *J*_H,H_ = 8.8, 1H, Ar-H), 7.73 (dd, *J*_H,H_ = 8.9; 2.5, 1H, Ar-H), 7.57 (s, 1H, triazole-H), 7.41–7.33 (m, 1H, Ar-H), 7.27–7.18 (m, 2H, Ar-H), 7.04 (dd, *J*_H,H_ = 8.4; 2.4, 1H, Ar-H), 6.99 (d, *J*_H,H_ = 8.9, 1H, Ar-H), 6.69 (d, *J*_H,H_ = 3.3, 1H, furyl-H), 6.37 (d, *J*_H,H_ = 3.3, 1H, furyl-H), 5.15 (s, 2H, -ArO-CH_2_-Ar), 4.37 (t, *J*_H,H_ = 5.6, 2H, -CH_2_-CH_2_-triazole), 3.91 (s, 2H, -N-CH_2_-furyl), 3.80 (s, 2H, -N-CH_2_-triazole), 3.40 (t, *J*_H,H_ = 6.4, 2H, CH_2_-CH_2_-SO_2_), 3.26 (t, *J*_H,H_ = 6.5, 2H, -N-CH_2_-CH_2_-), 2.98 (s, 3H, -SO_2_CH_3_), 2–23–2.11 (bs, 4H, triazole-CH_2_-CH_2_-CH_2_-C_cluster_), 1.92 (s, 3H, C_cluster_-CH_3_). ^13^C{^1^H}-NMR (100.77 MHz, CDCl_3_) δ: 164.5, 161.3, 157.9, 154.7, 152.9, 150.9, 150.8, 149.3, 144.6, 139.2, 132.7, 130.1, 128.8, 128.6, 128.4, 125.1, 123.0, 122.6, 122.4, 115.7, 114.9, 114.6, 114.2, 113.7, 111.7, 106.8, 74.8, 70.4, 52.3, 49.3, 49.0, 48.3, 46.7, 42.0, 33.4, 30.2, 24.3. ^11^B{^1^H}-NMR (96.29 MHz, CDCl_3_) δ: −4.4 (1B), −5.9 (1B), −10.6 (8B). Anal. calcd. for: C_38_H_47_B_10_ClFN_7_O_4_S: C, 53.04; H, 5.51; N, 11.40. Found: C, 53.22; H, 5.88, N, 11.02.

#### 2.4.3. Hybrid **20**

Yellow solid (188 mg, 79%). ^1^H{^11^B}-NMR (400 MHz, CDCl_3_) δ: 8.70 (s, 1H, pyrimidine-H), 8.66 (s, 1H, -NH) 8.53 (s, 1H, Ar-H), 7.94 (m, 2H, Ar-H), 7.85 (d, *J*_H,H_ = 8.7, 1H, Ar-H), 7.74 (dd, *J*_H,H_ = 8.8; 2.5, 1H, Ar-H), 7.51 (s, 1H, triazole-H), 7.41–7.34 (m, 1H, Ar-H), 7.27–7.18 (m, 2H, Ar-H), 7.05 (dd, *J*_H,H_ = 8.6; 1.8, 1H, Ar-H), 7.00 (d, *J*_H,H_ = 8.9, 1H, Ar-H), 6.70 (d, *J*_H,H_ = 3.2, 1H, furyl-H), 6.37 (d, *J*_H,H_ = 3.2, 1H, furyl-H), 5.16 (s, 2H, -ArO-CH_2_-Ar), 4.27 (t, *J*_H,H_ = 5.6, 2H, -CH_2_-CH_2_-triazole) 3.91 (s, 2H, -N-CH_2_-furyl), 3.81 (s, 2H, -N-CH_2_-triazole), 3.43 (dd, *J*_H,H_ = 19.5; 13.2, 2H, CH_2_-CH_2_-SO_2_), 3.27 (dt, *J*_H,H_ = 17.9; 11.5, 2H, -N-CH_2_-CH_2_-), 2.99 (s, 3H, -SO_2_CH_3_), 2.92 (s, 1H, C_cluster_-H ) 1.97 (bs, 4H, triazole-CH_2_-CH_2_-CH_2_-C_cluster_). ^13^C{^1^H}-NMR (100.77 MHz, CDCl_3_) δ: 164.7, 161.4, 158.0, 154.9, 153.1, 150.9, 149.5, 144.8, 139.3, 132.8, 130.2, 130.1, 129.0, 128.7, 128.5, 125.3, 123.2, 122.5, 115.9, 115.7, 115.0, 114.7, 114.3, 113.9, 111.8, 106.9, 74.5, 70.5, 55.1, 52.4, 49.4, 49.2, 48.3, 46.8, 42.1, 33.6, 30.3. ^11^B{^1^H}-NMR (96.29 MHz, CDCl_3_) δ: −4.6 (1B), −11.1 (5B), −13.8 (2B), −15.7 (2B Anal. calcd. for: C_37_H_45_B_10_ClFN_7_O_4_S: C, 53.50; H, 5.36; N, 11.58. Found: C, 52.62; H, 5.71, N, 11.21.

#### 2.4.4. Hybrid **21**

Yellow solid (257 mg, 88%). ^1^H{^11^B}-NMR (400 MHz, CDCl_3_) δ: 8.69 (bs, 2H, pyrimidine-H and -NH), 8.54 (s, 1H, Ar-H), 7.94 (dd, *J*_H,H_= 6.2; 1.9, 2H, Ar-H), 7.85 (d, *J*_H,H_ = 8.7, 1H, Ar-H), 7.73 (dd, *J*_H,H_ = 8.8; 2.4, 1H, Ar-H), 7.51 (s, 1H, triazole-H), 7.41–7.33 (m, 1H, Ar-H), 7.26–7.21 (m, 2H, Ar-H), 7.04 (dd, *J*_H,H_ = 8.7; 1.9, 1H, Ar-H), 6.99 (d, *J*_H,H_ = 8.8, 1H, Ar-H), 6.69 (d, *J*_H,H_ = 3.3, 1H, furyl-H), 6.37 (d, *J*_H,H_ = 3.2, 1H, furyl-H), 5.16 (s, 2H, -ArO-CH_2_-Ar), 4.27 (t, *J*_H,H_ = 5.6, 2H, -CH_2_-CH_2_-triazole), 3.91 (s, 2H, -N-CH_2_-furyl), 3.81 (s, 2H, -N-CH_2_-triazole), 3.41 (t, *J*_H,H_ = 6.4, 2H, CH_2_-CH_2_-SO_2_), 3.33 (t, *J*_H,H_ = 6.5 Hz, 2H, -N-CH_2_-CH_2_-), 2.98 (s, 3H, -SO_2_CH_3_), 1.95 (bs, 4H, triazole-CH_2_-CH_2_-CH_2_-C_cluster_-, 1.67 (s, 3H, C_cluster_-CH_3_). ^13^C{^1^H}-NMR (100.77 MHz, CDCl_3_) δ: 164.7, 161.4, 158.0, 154.8, 153.1, 151.0, 149.5, 144.7, 139.3, 132.8, 130.2, 130.1, 128.9, 128.7, 128.5, 125.2, 123.1, 122.5, 115.9, 115.7, 115.0, 114.7, 114.3, 113.9, 111.8, 106.9, 74.7, 70.8, 70.5, 52.4, 49.4, 49.1, 48.4, 46.8, 42.1, 33.5, 30.3, 24.4. ^11^B{^1^H}-NMR (96.29 MHz, CDCl3) δ: −4.8 (1B), −7.2 (1B), −9.2 (6B), −11.6 (2B). Anal. calcd. for C_38_H_47_B_10_ClFN_7_O_4_S: C, 53.04; H, 5.51; N, 11.40. Found: C, 53.35; H, 5.63, N, 11.77.

#### 2.4.5. Hybrid **22**

Orange solid (142 mg, 73%). ^1^H{^11^B}-NMR (400 MHz, CO(CD_3_)_2_) δ: 9.20 (bs, 1H, -NH), 8.69 (s, 1H, pyrimidine-H), 8.60 (bs, 1H, Ar-H), 8.18–8.15 (m, 2H, Ar-H), 8.06 (s, 1H, Ar-H), 7.84 (d, *J*_H,H_ = 8.1, 1H, Ar-H), 7.88 (dd, *J*_H,H_ = 8.9; 2.2, 1H, Ar-H), 7.53–7.43 (m, 1H, Ar-H), 7.40 (s, 1H, triazole-H), 7.38–7.31 (m, 1H, Ar-H), 7.23 (d, *J*_H,H_ = 8.9, 1H, Ar-H), 7.15–7.09 (m, 1H, Ar-H), 6.95 (d, *J*_H,H_ = 2.6, 1H, furyl-H), 6.55 (d, *J*_H,H_ = 2.3, 1H, furyl-H), 5.29 (s, 2H, -ArO-CH_2_-Ar), 4.60 (t, *J*_H,H_ = 5.6, 2H, -CH_2_-CH_2_-triazole), 4.22 (bs, 4H, C_cluster_-H), 3.95 (s, 2H, -N-CH_2_-furyl), 3.93–3.87 (m, 2H, -CH_2_-CH_2_-triazole), 3.66–3.59 (m, 2H, -CH_2_-CH_2_-SO_2_), 3–58–3.52 (m, 2H, -N-CH_2_-CH_2_-), 3.45–3.35 (m, 6H, -CH_2_-CH_2_-C_cluster_ and -CH_2_-CH_2_-SO_2_) 3.09 (s, 3H, -SO_2_CH_3_). ^13^C{^1^H}-NMR (100.77 MHz, CO(CD_3_)_2_) δ: 164.5 161.3, 157.8, 152.5, 150.6, 144.6, 140.1, 140.0, 133.2, 130.4, 130.3, 129.1, 128.8, 128.5, 124.5, 123.1, 122.2, 122.1, 116.1, 114.6, 114.3, 114.3, 114.1, 113.8, 111.8, 107.7, 71.9 (2C), 69.9, 69.2 (2C), 68.5 (2C), 53.8 (2C), 52.3, 50.1, 48.1, 46.5, 41.6. ^11^B{^1^H}-NMR (96.29 MHz, CO(CD_3_)_2_) δ: 24.7 (1B), 6.0 (1B), 1.6 (1B), −1.3 (1B), −3.1 (2B), −6.1 (6B), −16.0 (2B), −19.0 (3B), −27.5 (1B). MALDI-TOF-MS: *m/z* calcd. for C_40_H_57_B_18_ClCoFN_7_O_6_S: 1074.48. Found: 1072.7446. Anal. calcd.: C: 44.82; H: 5.36; N: 9.15. Found: C: 44.61; H: 5.90; N: 9.27.

#### 2.4.6. Bioisoster 23

Yellow solid (69 mg, 91%). ^1^H-NMR (400 MHz, CDCl_3_) δ: 8.74 (s, 1H, pyrimidine-H), 8.69 (s, 1H, Ar-H), 8.56 (bs, 1H, -NH), 7.95–7.91 (m, 1H, Ar-H), 7.90 (d, *J*_H,H_ = 1.4, 1H, Ar-H), 7.84 (d, *J*_H,H_ = 8.4, 1H, Ar-H), 7.72 (dd, *J*_H,H_ = 8.9; 2.5, 1H, Ar-H), 7.51 (s, 1H, triazole-H), 7.41–7.29 (m, 2H, Ar-H), 7.27–7.18 (m, 4H, Ar-H), 7.14 (d, *J*_H,H_ = 7.2, 2H, Ar-H), 7.03 (t, *J*_H,H_ = 8.4, 1H, Ar-H), 6.96 (d, *J*_H,H_ = 8.9, 1H, Ar-H), 6.68 (d, *J*_H,H_ = 3.2, 1H, furyl-H), 6.36 (d, *J*_H,H_ = 3.2, 1H, furyl-H), 5.12 (s, 2H, -ArO-CH_2_-Ar), 4.35 (t, *J*_H,H_ = 7.1, 2H, -CH_2_-CH_2_-triazole), 3.91 (s, 2H, -N-CH_2_-furyl), 3.80 (s, 2H, -N-CH_2_-triazole), 3.40 (t, *J*_H,H_ = 6.7, 2H, CH_2_-CH_2_-SO_2_), 3.27 (t, *J*_H,H_ = 6.9, 2H, -N-CH_2_-CH_2_-), 2.98 (s, 3H, -SO_2_CH_3_), 2.64 (t, *J*_H,H_= 7.5, 2H, -CH_2_-CH_2_-Ar), 2.40–2.05 (m, 2H, -CH_2_-CH_2_-Ar). ^13^C{^1^H}-NMR (100.77 MHz, CDCl_3_) δ: 164.7, 164.6, 161.4, 158.1, 154.8, 152.9, 151.1, 150.9, 149.4, 144.3, 139.9, 139.2, 132.8, 130.2, 130.1, 128.8, 128.6, 128.5, 128.4, 126.4, 125.3, 123.1, 122.9, 122.5, 115.9, 115.7, 114.9, 114.7, 114.2, 113.8, 111.8, 106.9, 70.4, 52.4, 49.7, 49.1, 48.4, 46.8, 42.1, 32.5, 31.6. Anal. calcd. for C_41_H_39_ClFN_7_O_4_S: C, 63.11; H, 5.04; N, 12.57. Found: C, 63.51; H, 5.12, N, 12.21.

### 2.5. Biology

#### 2.5.1. Tumor Cells

EGFR-overexpressing cell lines, glioma cell lines U87 MG (ATCC HTB-14) and C6 (ATCC CCL-107), and colorectal adenocarcinoma cell line HT-29 (ATCC^®^ HTB-38) were obtained from ATCC culture collection (Virginia City, NV, USA). F98 rat glioma cells, histologically characterized as an anaplastic astrocytoma, were a kind gift from Dr. Rolf Barth (Dept. of Pathology, The Ohio State University, Columbus, OH, USA). Cells were cultured in Dulbecco modified Eagle’s medium high glucose (4.5 g/L) with stable glutamine (3.97 mM) (DMEM) supplemented with 10% heat-inactivated fetal bovine serum (i-FBS) and 1% of a solution containing penicillin-streptomycin (PS-B) at 37 °C in a humid atmosphere containing 5% CO_2_ concentration. Culturing materials were purchased from Capricorn Scientific. In order to obtain i-FBS, original FBS was heated at 60 °C for 30 min.

#### 2.5.2. Glia Primary Cell Culture

A mixed glia primary cell culture was obtained from cerebral cortex of newborn mice as previously described with minor modifications [29]. Cerebral cortex of 3–4 newborn mice were dissected with the meninges carefully removed. Cerebral cortex were mechanically chopped and then enzymatically dissociated in 0.25% trypsin for 10 min at 37 °C. Fetal bovine serum (FBS) 10% in DMEM was then added to halt trypsin digestion. Repetitive pipetting thoroughly disaggregated the tissue, which was then strained through an 80 µm mesh and spun down. The pellet was re-suspended in glia culture medium and plated in T25 culture flasks. Glia culture medium composition is as follows: DMEM supplemented with 10% FBS, HEPES buffer (3.6 g/mL), penicillin (100 IU/mL), and streptomycin (100 µg/mL). Culture medium was replaced every 48 h until 90% confluence was reached.

### 2.6. Study Approval

All procedures using laboratory animals were performed in accordance with the national and international guidelines and were approved by the Institutional Animal Committee for animal experimentation. All experimental procedures were approved by the Ethical Committee for the use of Experimental Animals (CEUA) of the Institut Pasteur de Montevideo, Uruguay (CEUA Approved protocol: #003-17 to Dr. Emiliano Trias), and under the current ethical regulations of the Uruguayan Law 18.611 for animal experimentation that follows the Guide for the Care and Use of Laboratory Animals of the National Institutes of Health [30].

### 2.7. Cytotoxicity Assays

Cytotoxicity assays were performed as previously described with minor modifications [18]. EGFR-overexpressing cells or glia cells were seeded in 96-well plates (7 × 10^3^ – 1 × 10^4^ cells/well depending on the cell line for tumor cells and 2 × 10^4^ cells/well for glia cells) in 100 μL final volume of growing medium and were allowed to grow for 24 h. After that, 125 μL of fresh culture medium was added and the cells were allowed to grow for additional 24 h. Then, 25 μL of a solution 10× the desired final concentration of the tested compounds in culture medium was added to the culturing media. Cells were further incubated for 24 h. Afterwards, culture medium was removed and cells were washed twice with 200 μL of PBS. Cells were then fixed with 50 μL of ice-cold trichloroacetic acid for 1 h at 4 °C. Next, the plates were washed five times in distilled water and allowed to dry at room temperature. Sulphorhodamine B (SRB) solution (50 μL, 0.4 *w*/*v* in acetic acid 1% *v*/*v*) was added to each well of the dried 96-well plates [31]. Staining was performed at room temperature for 30 min. The SRB solution and unbound dye were removed by washing the plates quickly with aqueous solution of acetic acid (1%, *v*/*v*) at least five times (until excess dye was fully removed). The washed plates were allowed to dry at room temperature for at least 24 h. Finally, the bound SRB was solubilized by adding Tris Base buffer (pH 10, 10 mM, 100 μL) to each well, and the resulting solution was shaken for 5 min on a shaker platform. The optical density (OD) of each well solution was read in a 96-well plate reader at λ = 540 nm. The OD of SRB solution in each well is directly proportional to the cell number. Cell viability percentage was calculated according to the following equation: CV% = (A_540 nm_ – B)/(C – B), where CV% stands for cell viability percentage, A_540 nm_ corresponds to OD of a particular well, B is the OD of untreated wells with no cells seeded onto them, and C is the OD of control wells treated only with 1% of DMSO. CV% values were plotted against compound concentration, and the IC_50_ values were determined.

### 2.8. Kinase Enzymatic Assays

The enzymatic activity of hybrid **19** and **Lap** against EGFR was determined using ADP-Glo™ kinase Assay system from Promega Corporation (Catalog number V3831, Fitchburg, WI, USA). The experiments were conducted according to the manufacturer instructions. Briefly, an EGFR kinase reaction mixture of 10 μL was carried out by addition of 2 μL of the compound dilution, 3 μL of the enzyme dilution, and 5 μL of the ATP/substrate solution to get a final concentration of 50 μM ATP, 0.2 g/mL of Poly(Glu, Tyr) substrate and 5 mg/mL of enzyme. Each reaction was performed by triplicate on a 384-well plate and incubated for a duration of 1 h at room temperature. Final compound concentrations in the kinase reaction were 100, 10, 1, 0.1, 0.01, 0.001, and 0.0001 μM. After the reaction time, 5 μL of ADP-Glo™ was added and allowed to react during 40 min. Finally, 10 μL of Kinase Detection Reagent was added and incubated for additional 40 min. Afterwards, luminescence was detected on a BioTek^®^ FLx800 Multi-Detection Microplate Reader (Integration time 0.5–1 s). Curve fitting and data presentations were performed using GraphPad Prism version 8.0.

### 2.9. Flow Cytometry Analysis

In order to study cell death mechanism (namely apoptosis or necrosis) triggered by exposure to hybrid **19** or **Lap** of glia and U87 MG cells, phosphatidylserine exposure was measured by flow cytometry using Annexin-V staining. Glia cells (6 × 10^4^ cells) were seeded into p60 dishes and were allowed to grow until 80% confluence. Next, 1 × 10^5^ U87 MG stained cells with PKH26 dye were seeded onto a p60 dish containing a growing glia primary cell culture. After a 48 h incubation in fresh growing medium, the p60 dishes were treated with IC_50_ dose of hybrid **19** or **Lap** at 10 µM or 100 µM. A p60 dish treated with 1% DMSO served as control. Treated cells were incubated for 24 h. Then, cells were harvested with trypsin (0.05%, supplemented with EDTA, 0.38 mg/mL) and centrifuged at 250 g speed. The resulting pellet was resuspended in an appropriate volume of Annexin binding buffer (0.01 M Hepes pH 7.4, 0.14 M NaCl, and 2.5 mM CaCl_2_) to get a cell suspension of 1 × 10^6^ cells per mL. After cell counts, samples were divided, and cells alone and isotype-matched control samples were generated to control for nonspecific binding of antibodies and for autofluorescence. An Annexin V-FITC antibody solution (catalog number: A13199) was used at a 1:20 concentration. After 30 min of incubation with the aforementioned antibody at 4 °C, samples were incubated with DAPI at a 1:5000 concentration and were immediately analyzed by flow cytometry. To perform the analysis, cells were first gated for PKH26 in order to distinguish U87 MG cells from glia cells. In each subpopulation of cells, the following markers were used to define four different populations: viable cells (Annexin-V^−^/DAPI^−^), necrotic cells (Annexin-V^−^/DAPI^+^), early apoptotic cells (Annexin-V^+^/DAPI^−^), and finally late apoptotic cells (Annexin-V^+^/DAPI^+^). Samples were acquired using FACSAria Fusion flow cytometer (Biosense, Irwindale, CA, USA). and BD FACSDiva™ software (Biosense). Three independent flow cytometry experiments were done, except for **Lap,** evaluated at 100 μM, which was included in only one of these experiments. Data obtained from these experiments were analyzed using FlowJo software (FlowJo LLC, Ashland, OR, USA).

### 2.10. Statistical Analysis

Statistical analysis was done using GraphPad Prism 8.0 software (GraphPad, LaJolla, CA, USA). For statistical analysis of flow cytometry data, unpaired T Test was performed to determine whether each population present on the dotplot DAPI vs. Annexin-V-FITC of each experimental condition was different from control populations. A *p*-value < 0.05 was considered significant.

### 2.11. Fluorescence Microscopy Analysis

Fluorescence microscopy studies were performed on in vitro co-cultures comprising U87 MG cells plated along with glia cells. Co-cultures were set by seeding glia cells (3 × 10^4^ cells) into p35 dishes. Glia cells were allowed to grow until 80% confluence. Next, 5 × 10^4^ U87 MG stained cells with PKH26 dye were seeded onto a p35 dish containing growing glia cells. After a 48 h-incubation in fresh growing medium, the p60 dishes were treated with IC_50_ dose of hybrid **19** or **Lap** at 10 µM or 100 µM. A p60 dish treated with 1% DMSO served as a negative control. Treated cells were incubated for further 24 h. After that, treated co-cultures were fixed for 20 min at 4 °C with PFA and washed 2 times with PBS. Then, samples were permeabilized for 10 min at room temperature with 0.1% Triton X-100 in PBS, passed through washing PBS, blocked with 5% BSA:PBS for 1 h at room temperature, and incubated overnight in a solution of 1% BSA:PBS containing the primary antibodies and DAPI at 4 °C. After washing, treated cells were incubated in 1:500-diluted secondary antibodies during 2 h at room temperature. The following antibodies were used for immunofluorescence staining: primary antibodies 1:400 mouse monoclonal anti-GFAP (Sigma-Aldrich, Darmstadt, Germany) and 1:50 rat monoclonal anti M2, and secondary antibodies conjugated to AlexaFluor 633 AlexaFluor 488 (Invitrogen, Carlsbad, CA, USA), respectively. DAPI was used at a 1:1000 dilution. Antibodies were detected by confocal microscopy using a confocal Zeiss LSM 800 microscope (Jena, Germany).

### 2.12. In Vitro BNCT Experiments

#### 2.12.1. Determination of Intracellular Boron by Inductively Coupled Plasma Optical Emission Spectroscopy (ICP−OES).

Exponentially growing cells (F98 or HT-29) were seeded in plates of 24 wells and incubated with hybrid **19** (10 μM, dissolved in DMSO), **22** (10 μM, dissolved in DMSO) or ^10^B-**BPA** (0.925 mM, diluting with water the stock solution to a final volume of 500 μL). The stock solution of ^10^B-**BPA** was prepared at a concentration of 30 mg ^10^B-**BPA**-fructose (99% ^10^B enriched, L-isomer) (Glyconix Corp, Raleigh, NC, USA) per mL (0.14 M) as follows: ^10^B-**BPA** was combined in water with a 10% molar excess of fructose, adjusting the pH to 9.5–10.0 with aqueous solution of NaOH, and the resulted mixture was stirred until all solids were dissolved; finally, the pH was readjusted to 7.4 with aqueous solution of HCl [32]. After varying times of incubation (1, 2, and 4 h for F98 and 1, 2, 4, 6, 24, and 48 h for HT-29), the cells were washed twice with PBS 1× at 4 °C. The pellets were digested with formic acid (500 μL). An aliquot of 250 μL was diluted to 1.0 mL with an aqueous solution containing 1 mg/L of Y (0.75 mL) as internal standards. The boron uptake was measured by ICP–OES. Analytical and internal standard lines (in nm) were as follows: B: 249.677 and Y: 371.029. Matrix-matched standard solutions containing the internal standard elements and boron between 0.05 and 0.75 mg/mL were employed for daily calibration [33,34,35]. An internal measurement control was also used, with each certain number of samples adding a sample of known concentration (QC). The other aliquot of 250 μL was dried and re-suspended in aqueous solution of NaOH (0.3 N), and total proteins were determined by Lowry method. The boron amount was referred to the total proteins.

#### 2.12.2. Neutron-Irradiation Procedures

Cells were irradiated at the thermal column of the RA-3 reactor, a 9 MW nuclear reactor located in Ezeiza (Argentina), where a highly thermalized and homogenous irradiation field is available. Thermal flux was near to (1.0 ± 0.1) × 10^10^ neutrons cm^−2^ s^−1^; the cadmium ratio was 4100 for gold foils, which allows neglecting fast neutron dose; and the gamma dose rate was approximately (6.0 ± 0.2) Gy h^−1^. Total dose was obtained by adding partial doses coming from photons, nitrogen capture (a 3.5% wt of nitrogen content is assumed), and boron capture. Before each irradiation, neutron flux at the irradiation position was checked using calibrated Rh-SPND detectors in a system that mimics the configuration that will be used (96-well plates with no cells inside), while, simultaneously, signal from a boron-coated ionization chamber was used as a monitor. Based on this measurement, irradiation times were calculated in order to deliver a dose of 1 and 2 Gy with an estimated uncertainty of 10% [36]. The dosimetry, for each treatment, the time, the fluence and each dose component of the total physical absorbed dose of the neutron beam without and with boron has been studied. The intracellular boron concentration at the time of irradiation was assumed uniformly distributed inside and outside the cells for the dosimetric calculations.

#### 2.12.3. Irradiation of Hybrid **19**-, **22**- or **BPA**-Treated HT-29 Cells Assays

HT-29 cells were seeded at 2500 cells/well in 96-well plates. Eight different wells were seeded per treatment. The cells were then treated with 10.0 ppm of ^10^B of hybrids **19** or **22** (dissolved in DMSO) or ^10^B-**BPA** (dissolved in water) for 1 h after plating. Irradiated DMSO (5%)-treated cells and non-irradiated compound-treated cells served as controls. After this time, the cells were irradiated with thermal neutrons as described above.

#### 2.12.4. Cell Surviving Assay

After irradiation, the medium was changed and the cells were cultured at 37 °C for 10 days. Afterwards, 20 μL of vital dye 3-(4,5-dimethylthiazol-2-yl)-2,5-diphenyltetrazolium bromide (MTT, Sigma 128, 0.5%, *w*/*v* in PBS) was added to the culture medium, and after 4 h of incubation at 37 °C, absorbance at 540 nm was observed. Results are expressed as percentage of untreated controls.

## 3. Results and Discussion

### 3.1. Design and Synthesis of Hybrids Carboranyl-Decorated Lapatinib-Scaffold

The following two structural features are responsible for effective **Lap** EGFR interaction [37]: i) the quinazoline ring, via its nitrogens that establish hydrogen bonds to Met769 and Thr830, and sandwiching between Ala719 and Leu820; and ii) the fluorobenzyloxyphenylamino moiety that makes hydrophobic interactions in the back of the ATP binding site. On the other hand, the methylsulfonylethylamino group is positioned at the solvent interface without significant interactions with the protein, establishing poor interaction to Asp776. For these reasons and considering the structural requirements, for the new designed hybrids we selected the solvent-exposed ethylamino-moiety to bind the high boron content cages using a polar linker, i.e., [1,2,3]triazolyl moiety [20] (Figure 1). Due to the C_cluster_-H and B-H vertices, boron clusters could establish special hydrogen and dihydrogen bonds, such as C-H···X [38] and BH···H-X (X = N, C, O, and S), as well as BH···π, C-H···π hydrogen bonds [39,40], and C–H···Halogen interactions (Halogen = F, I [41,42]); three types of clusters were incorporated into the **Lap** scaffold, the neutral *closo*-carboranes—i.e., 1,2-dicarba-*closo*-dodecarborane and 1,7-dicarba-*closo*-dodecarborane—and the anionic cobaltabis(dicarbollide).

Consequently, our synthetic approach for the new hybrids starting from **Lap** is outlined in Scheme 1. **Lap** was selectively propargylated in the aliphatic amine in order to subsequently apply 1,3-dipolar cycloadditions with carboranyl-containing azides **8**, **9**, **12**, **13**, and **18**. The desired hybrids, **18–22**, connected by the polar [1,2,3]triazolyl moiety, were obtained in good to excellent yields. To confirm the relevance of carboranyl moiety in the displayed bioactivity (see below) we prepared the bioisoster [43,44] phenyl-analogue **23** (Scheme 1) of hybrids **18** and **20** using azide intermediate **17**. The intermediates and final compounds were structurally characterized in terms of ^1^H-, ^11^B{^1^H}-, ^13^C{^1^H}-NMR, and UV-spectroscopies, mass spectrometry, and elemental microanalyses (C, H, N, and S).

### 3.2. Biological Studies

#### 3.2.1. In Vitro Cytotoxicity Studies

To address the issue of human therapy, in vitro cytotoxicities of the hybrids **18**–**22** and the bioisoster **23** were determined on TKRs-overexpressing cells [18]: *Homo sapiens* colorectal adenocarcinoma HT-29 and brain glioblastoma U87 MG. For further animal in vivo experiments, *Rattus norvegicus* brain glioma C6 were also included in this study (Table 1). Compared to parent compound **Lap**, the hybrids resulted poorly active against HT-29 cells, being the most cytotoxic the Cobaltabis(dicarbollide) derivative **22** and the 1,2-dicarba-*closo*-dodecarborane **18**. However, against glioma cells, a marked cytotoxic behavior was evidenced mainly for compound **19**, resulting in 4.6- and over 10-times more active than the parent compound, **Lap**, against C6 and U87 MG cells, respectively. Additionally, hybrids **18**, **21**, and **22** were more active than **Lap** against U87 MG glioma cells. These results encouraged us to evaluate hybrid **19** against normal glial cells isolated from neonatal cortex to determine if this compound selectively killed tumor cells, while having no cytotoxic effects on normal glial cells (Table 1). The results showed that hybrid **19** displays tumor-cell selectivity with excellent selectivity indexes compared to the parent compound **Lap**. The benzyl-derivative **23** was less active than the corresponding bioisoster carborane hybrids **18** and **20** in all the studied cellular systems, confirming the relevance of the boron cages in the biological behaviors.

#### 3.2.2. In Vitro Inhibition of EGRF by Hybrid **19**

With EGFR being one of the bio-systems inhibited by **Lap**, we analyzed the inhibition of the most cellular-cytotoxic hybrid, **19**, on this isolated system, comparing to **Lap** inhibition ability. The hybrid **19** was 10 times less potent than **Lap** against EGFR (IC_50,**19**_ ~3.0 μM, IC_50,**Lap**_ ~0.3 μM, Appendix A). However, it should be noted that **Lap**, which is not a typical Type II inhibitor because of its unusual displacement of the α-C helix, resulted in in vitro high-throughput competition-binding assay (KINOMEscan, Appendix A), in an extremely selective inhibitor, against EGFR and Erb2. Thus, *closo*-carboranyl derivative **19** could be acting on glioblastoma cells, inhibiting other tyrosine kinase proteins, or also through different mechanisms of action.

#### 3.2.3. Effect of Hybrid **19** on Simulated Tumor Environment

In order to study the effect of hybrid **19** more deeply, we simulated the cellular environment of a developing brain tumor co-culturing U87 MG cells, stained with PKH26 dye, together with neonatal murine astrocytes. Co-cultured cells were incubated for 24 h with hybrid **19** at U87 MG IC_50_ dose, **Lap** at 10 μM (near to IC_50,astrocyte_ dose), and 100 μM (lower than IC_50,U87 MG_ dose), or dimethylsulfoxide (DMSO, 1%) as negative control. Confocal microscopy revealed very different behaviors among the studied compounds (Figure 2, see merge images). As expected, PKH26 dye showed a decrease of U87 MG population upon **19**-treatment compared to control vehicle-treated cells. However, according to astrocytic markers M2 for astrocytes membrane and GFAP for astrocytes activation [45,46], the population of tumor-associated astrocytes was observed mainly in the hybrid **19**-incubated cells and in the control, while it was not present in the co-culture treated with **Lap** at the higher dose. Moreover, when comparing GFAP staining of lower dose **Lap**-treated cells and control cells, it was possible to observe that **Lap** made astrocytes become reactive, as indicated by increased GFAP expression, but hybrid **19** had a similar effect to that of the control (see merge images). Finally, DAPI nuclei staining showed on hybrid **19**-treatment, like in **Lap** at the lower dose, some degree of chromatin condensation, which could be associated with the onset of apoptosis (Figure 2, zoomed-in inset).

For this reason, the co-cultured systems were phosphatidylserine-exposure analyzed by flow cytometry. The flow cytometry analysis showed that hybrid **19** produced cellular death of U87 MG tumoral cells by apoptosis and necrosis. Compared to untreated control, hybrid **19** significantly increased the percentage of cells undergoing late apoptosis from ~0.6% to ~35% and necrosis from 0.05% to ~7% (Figure 3). Similarly, associated astrocytes mainly died by apoptosis as a result of hybrid **19** exposure (56% of late apoptosis), with nearly 3% of necrosis. **Lap** displayed very different death behavior according to the assayed dose. At 100 μM, a lower dose than its IC_50_ against U87 MG, **Lap** promoted on U87 MG mainly late apoptosis (~95%). However, at 10 μM (a dose near to the astrocytes-cytotoxic dose), **Lap** promoted late apoptosis, ~42% on associated astrocytes (Figure 3).

### 3.3. In Vitro BNCT Studies

For these studies, we selected two of the most relevant hybrids, i.e., **19** and **22**. On the one hand, the *closo*-carboranyl derivative **19** displayed moderate cytotoxicity against HT-29 cell line, which displays a key factor in order to fulfill the requirements for successful BNCT. Additionally, **19** showed very interesting biological behaviors against glioblastoma cells; consequently, combined with neutron irradiation, it could be used in malignant gliomas. On the other hand, the derivative **22** displayed moderate activity against colorectal cells, optimum for BNCT treatments, and it bears a water-solubility-enhancing moiety, the metallacarboranyl group, that improves its drug-like properties. Besides, it is able to deliver larger amounts of ^10^B atoms to cells (18 times more B per **22**-molecule than **BPA**, 1.8 times more B per **22**-molecule than **19**, 1.5 times more B per **22**-molecule than **BSH**).

#### 3.3.1. Boron Cellular Accumulation

First, we analyzed the presence of boron promoted by **19** and **22** into two different cellular systems, i.e., HT-29 and *Rattus norvegicus* brain glioblastoma F98 cells to address further in vivo animal BNCT studies. Among the different ways to calculate the boron cellular concentration (g of boron/g of tumor tissue, number of boron atoms/number cells [7,8,9] or g of boron/mg of protein [47,48]) reported nowadays, the latest one has been chosen in this article. Boron accumulation as a result of **19**- and **22**-incubations, at 10 μM doses, was detected in HT-29 cells even after 48 h of treatments (values near to 0.5 μg of boron/mg of protein content for both compounds, Figure 4a) with the highest accumulations within the first hour, i.e., 4.5 μg of boron/mg of protein content for **19** and 3.2 μg of boron/mg of protein content for **22**. According to our previous results [33], HT-29 accumulates 0.5 μg of boron/mg of protein content when it was incubated for 24 h with **BPA** but with a dose 90 times higher than that indicated above for **19** and **22**. In F98 **19** and **22** incubation, for 4 h and at 10 μM doses, ^10^B-**BPA** was used at 0.925 mM dose as the reference compound. After 2 h of incubation the amount of boron produced by hybrids **19** and **22** was very important, decreasing with time for *closo*-carborane **19** and maintaining constantly a high value for metallacarborane **22** (Figure 4b). Additionally, F98 boron accumulation due to **BPA** was lower than **19** and **22** accumulations at 1 and 2 h of incubation, and unlike these compounds, it significantly decreased with time as a result of efflux by extracellular amino acids exchange (Figure 4b) [49].

#### 3.3.2. Neutron Irradiation Treatments

The previous information encouraged us to perform in vitro neutron irradiation experiments with both hybrids. In this sense, the HT-29 cells were incubated for 1 h with hybrid **19** or **22** using ^10^B-**BPA** as reference, and doses of 1 and 2 Gy were delivered. The irradiation times to deliver these doses, 1 and 2 Gy, were determined by dosimetric studies. Table 2 shows the dosimetry, for each treatment, the time, the fluence and each dose component of the total physical absorbed dose of the neutron beam without and with boron. All the studied compounds were incubated at doses equivalents to 10.0 ppm of ^10^B. The effect of in vitro BNCT displayed by hybrids **19** and **22** was very interesting, where a marked and statistically significant decrease was observed with respect to the irradiation with neutrons alone in the surviving cell fraction at 1 and 2 Gy-irradiated groups (Figure 5). Contrarily, at 1 and 2 Gy-irradiation and at 10.0 ppm of ^10^B, **BPA** produced a surviving cell fraction statistically significantly higher than hybrids **19** and **22** (Figure 5). The **BPA** effect was statistically significant, with respect to irradiation with neutrons alone, at 2-Gy of dose. Hybrid **19** was at least 7 times more effective than **BPA,** and hybrid **22** was at least 6 times more effective than **BPA,** in the in vitro model of BNCT.

## 4. Conclusions

A series of hybrid compounds using the tyrosine kinase receptors inhibitor lapatinib, as structural scaffold, and *closo*-carboranyl and metallacarboranyl moieties, as decorators, were developed as potential anti-tumor agents by dual mechanisms of action. The novel hybrids were evaluated in a panel of tumoral cells that overexpress tyrosine kinase proteins. Additionally, two hybrids were studied in a BNCT in vitro model and demonstrated promising behaviors.

The overall attractive profile of **19** and **22** makes them interesting compounds for further development. In this regard, further assessment will focus to extend insights into the mechanism of action through KINOMEscan’s in vitro competition binding assay, the preparation of hybrids **19** and **22**
^10^B-enriched, and assessment of BNCT in vitro and in vivo efficacy.

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
