# Peer review of "Closo-Carboranyl- and Metallacarboranyl [1,2,3]triazolyl-Decorated Lapatinib-Scaffold for Cancer Therapy Combining Tyrosine Kinase Inhibition and Boron Neutron Capture Therapy"

_cells, 2020, doi:10.3390/cells9061408_

Round 1
Reviewer 1 Report
The manuscript by Couto et al describes the synthesis and a biological evaluation of conjugates between lapatinin, a TKR-inhibitor, and a boron cluster as BNCT agents. The work is quite well described, and the results are interesting.
I would suggest publishing the manuscript after some minor revisions:
A general point which should be addressed is related to the potential toxicity of the compounds used for BNCT treatment. As the BNCT agents are used in huge amount, they must be non-toxic. In the case of the use of boron agents which have some toxicity, much attention should be payed to the balance between the therapeutic dose and any other effect. Of course, if a compound has much better uptake by cells than BPA (or BSH) the dose may be highly reduced, but I feel that this aspect must be evaluated in vivo. I would suggest addressing the problem in this paper, which anyway takes in account an evaluation of the toxicity of the tested compounds.
The authors use as boron moiety 1,2-carboranes, 1,7-carboranes and a cosan: could the authors explain why they decide to use any of these boron clusters and if they found (also with reference to previous works) any general indication for a preferred boron cluster (or it is strongly dependent on cell line?)?
The authors use as measure of boron concentration, micrograms of boron/mg of protein; in other contexts, data are given as number of boron atoms /number cells. As the most used way is micrograms of boron/g of tumor tissue, can the authors give an evaluation of their boron concentration in the last format?
Some minor comments:
- Page 2 lines 51 and 52: I would suggest to define EGFR and ErbB2
- Page 2 lines 53: I would suggest to substitute “essential therapy” by “promising therapy”
- Page 4 lines 136: I cannot consider the evaporation of the solvent as a quenching, just say that the solvent was evaporated and the crude taken up in dichloromethane.
- In 1H NMR data, please state that BH hydrogens are omitted from description and integration. Moreover, the way in which the CH of carborane cage is described was initially quite confusing for me, it could be simpler; instead of B-Ccluster-H, I would suggest to indicate it as cluster C-H, (not mandatory). Finally, as the proton signals have been assigned, I presume that at least COSY spectra have been recorded, if so please add them to the supporting information.
- Page 8 line 165: please add some references about carborane cage CH hydrogen bonds, see e.g. Am. Chem. Soc. 2002, 124, 30, 8778.
- Page 12 Fig. 4a: did the authors have any explanation for the increased concentration at 6 hours after incubation?
- Finally, I found the English style of some sentences unclear, please improve
Author Response
Reviewer 1:
Reviewer (R): A general point which should be addressed is related to the potential toxicity of the compounds used for BNCT treatment. As the BNCT agents are used in huge amount, they must be non-toxic. In the case of the use of boron agents which have some toxicity, much attention should be payed to the balance between the therapeutic dose and any other effect. Of course, if a compound has much better uptake by cells than BPA (or BSH) the dose may be highly reduced, but I feel that this aspect must be evaluated in vivo. I would suggest addressing the problem in this paper, which anyway takes in account an evaluation of the toxicity of the tested compounds.
Answer (A): One of the hybrids, along with the parent compound (lapatinib), was evaluated against normal glial cells (see Table 1) finding lower level of cytotoxicity of the developed compound.
Additionally, our previous results (references 18, 20 in the manuscript) indicated that these clusters showed absence or lower cytotoxicity as well as absence of mutagenic effects on normal cells.
R: The authors use as boron moiety 1,2-carboranes, 1,7-carboranes and a cosan: could the authors explain why they decide to use any of these boron clusters and if they found (also with reference to previous works) any general indication for a preferred boron cluster (or it is strongly dependent on cell line?)?
A: In this manuscript, we report on a new family of bifunctional compounds that offering the possibility of dual action (drug-radiotherapy combinations) may result into significant clinical benefits. This new family of compounds combines two fragments: the drug fragment is a lapatinib group, which inhibits the tyrosine kinase receptor activity and, the radiotherapy fragment, which is an icosahedral boron cluster that provides high boron contents (a requirement by BNCT) to the molecules. This is the second family of compounds that the authors have synthesized; the first one (references 18-20 in this revised manuscript file) is based on the erlotinib + icosahedral boron clusters.
Our previous results (references 18-20 in this revised manuscript file) suggested to us to choose the highly stable and 3D-aromatic icosahedral clusters: two neutral dicarboranes ortho- and meta-C2B10H12 isomers and a monoanionic sandwich cobaltabis(dicarborane), [Co(C2B9H11)2]- (known as COSAN) to start the research of this lapatinib-boron cluster family of compounds reported in this amended article.
We chose two isomers of neutral dicarborane because the meta- isomer is more stable against nucleophilic agents and more lipophilic than the ortho- isomer. In addition, due to the different position of the carbon atoms into the cluster, the hydrogen atoms bonded exo-cluster at the Carbon and Boron vertices present different partial charges that make the ortho- isomer more acid than the meta- and the interactions of the B-H vertices with the receptor of the TKR enzyme are different between both isomers.
To emphasize that icosahedral boron clusters provide high boron contents to the molecules, we have made a minor change in one sentence of the introduction.
It was written in the previous version: “These bifunctional compounds, which exploited the TKR-interaction/inhibition ability plus the selective boron accumulation for BNCT process, would act as anticancer bimodal agents (chemo + radiotherapy) to result in significant clinical benefits: reducing the doses to get the same therapeutic effect while diminishing the side effects to the patient”.
It is written in this amended version: “These bifunctional compounds, which combine the TKR-interaction/inhibition ability plus the selective and high boron atoms loading capacity in cells for the BNCT process, would act as anticancer bimodal agents (chemo + radiotherapy) to result in significant clinical benefits: reducing the doses to get the same therapeutic effect while diminishing the side effects to the patient.”
R: The authors use as measure of boron concentration, micrograms of boron/mg of protein; in other contexts, data are given as number of boron atoms /number cells. As the most used way is micrograms of boron/g of tumor tissue, can the authors give an evaluation of their boron concentration in the last format?
A: In our laboratory we have developed a method to measure boron content by ICPAES from cells previously seeded in plates of 24 wells (Dagrosa et al Thyroid. 2002, 12, 7-12, reference 40 in the original version of the manuscript). As it was described, it consists on washing twice with PBS 1X after incubating the cells with boron compounds and then to digest the cells with a small volume (0.5 mL) of formic acid (the number of cells in each well is around 104). In this way, the total digestion of the cells is achieved, and an aliquot is taken to measure boron content and another aliquot to measure proteins by Lowry. This method allows us to evaluate the uptake of boron compounds using a small amount of each one of them.
On the other hand, although the total proteins represent around a 20 % of total cell weight, it is not possible to compare µg of boron by grams of proteins weight (ppm) with µg of boron by 104 cells (ppm). However, we can compare the different boron compounds analyzed (19 and 22) between them and with BPA.
R: Some minor comments:
- Page 2 lines 51 and 52: I would suggest to define EGFR and ErbB2
- Page 2 lines 53: I would suggest to substitute “essential therapy” by “promising therapy”
A: The referee’s suggestions have been taken into account in the revised manuscript file.
R: - Page 4 lines 136: I cannot consider the evaporation of the solvent as a quenching, just say that the solvent was evaporated and the crude taken up in dichloromethane.
A: We fully agree with the reviewer that we made a mistake describing the evaporation of the solvent as a quenching process. We have corrected this mistake in the revised manuscript file. Thanks to the referee for his/her comment.
R: Some minor comments:
- In 1H NMR data, please state that BH hydrogens are omitted from description and integration. Moreover, the way in which the CH of carborane cage is described was initially quite confusing for me, it could be simpler; instead of B-Ccluster-H, I would suggest to indicate it as cluster C-H, (not mandatory).
A: We agree with the referee and his/her recommendation has been taken into account throughout the full revised manuscript.
R: Some minor comments:
- Finally, as the proton signals have been assigned, I presume that at least COSY spectra have been recorded, if so please add them to the supporting information.
A: We did not acquire the COSY spectra because the skeleton of lapatinib fragment (where the H-H correlations are relevant) was assigned by acquiring the 1H NMR of this parent compound. Then, once the proton from the lapatinib fragment were assigned, the new incorporated C-protons from the two fragments’ connecting groups do not display correlations H-H (H-triazole-H and –N-CH2-triazole).
R: Some minor comments:
- Page 8 line 165: please add some references about carborane cage CH hydrogen bonds, see e.g. Am. Chem. Soc. 2002, 124, 30, 8778.
A: This reference dealing with the presence of Cc-H···O bond has been added (reference 38) in this revised manuscript file.
R: Some minor comments:
- Page 12 Fig. 4a: did the authors have any explanation for the increased concentration at 6 hours after incubation?
A: We are very interested on these compounds’ behaviour but, we do not have any answer to the referee's question yet. We plan experiments that may allow us to obtain answers about these phenomena. As the mechanism of cellular uptake of these hybrids is not described yet, we will study possible mechanisms of entry (endocytosis mediated by receptor or cassette (ABC) transporters) as well as of action of these boron compounds in future works.
The peak of boron after 6 hours of incubation could be due to a modulation of its transporter in the membrane, increasing the rate of uptake once incorporated. As an example of this, Chun-ling Dai et al (Cancer Res 2008; 68: 19) showed that lapatinib significantly enhanced the sensitivity to ABCB1 or ABCG2 substrates in cells expressing ATP-binding cassette (ABC) transporters.
R: Finally, I found the English style of some sentences unclear, please improve.
A: The English has been corrected by a chemist native English speaker.
Reviewer 2 Report
The article by Couto et al. describes new compounds for potential application in BNCT combined with receptor-based tumor inhibition. The authors did a great job synthesizing and testing novel potential boron compounds for BNCT. However, it is difficult to accept it in the present form. The following issues need to be figured out by the authors:
- If the authors put discussions in the Results section, they should change the section title to Results and Discussion, if it is allowed by the journal. Otherwise, the article should have a proper Discussion with the comparison of authors’ results with other related studies and listing the limitations, as a general rule.
- The use of the mentioned cell lines should be justified. In different experiments the authors used different cell lines. It is confusing and difficult to make any conclusions based on different experiments with different cell lines in each experiment. For example, U87 MG cells were used in flow cytometry analysis and fluorescence microscopy, whereas F98 or HT-29 cells were used in irradiation experiments. Why did the authors choose those lines, but not others? For example, “to address the issue of ….etc. we used the following cell lines….” etc. Or “these cells are planned to be used in further in vivo experiments”, etc.
- Proper information on the equipment/reagents origins should be provided, including cities and countries of the companies (as a general rule).
- The ethic committee approval or following the corresponding rules/protocols/conventions should be mentioned when using laboratory animals (here – primary cell culture preparation).
For example: “All manipulations with animals were carried out in compliance with research practices in accordance with the directives of the European Community (86/609 / EEC).”
- The type of the primary cell culture should be mentioned in the paragraph title (Primary glial cell culture, for example), as it is not clear from the title and the preparation protocol should be listed if there is any published previously. Is the preparation of the primary cells culture unique in your study or are there any previously published protocols that you used ore modified for that? If protocols exist, they should be mentioned and referenced.
- The protocol for the SRB-based cytotoxicity assay should be mentioned and put in references, as there are numerous publications of the described here cytotoxicity method. The chosen cytotoxicity assay should be justified, for example “to avoid direct interference of the compounds with the reagents…, etc.”, as many researchers use proliferation-based MTT, MTS, or other methods to show live cell rates, but here all cells are stained.
As an example, please refer to: Vichai, V.; Kirtikara, K. Sulforhodamine B colorimetric assay for cytotoxicity screening. Nat Protoc. 2006;1(3):1112-6. DOI: 10.1038/nprot.2006.179
- The protocol for BPA-fructose solution preparation should be cited, otherwise the authors need to prove that they used their own original protocol for BPA solution preparation, different from the generally accepted one, for example, published by Rossini et al. (2015) or others.
Rossini AE, Dagrosa MA, Portu A, Saint Martin G, Thorp S, Casal M, Navarro A, Juvenal GJ, Pisarev MA. Assessment of biological effectiveness of boron neutron capture therapy in primary and metastatic melanoma cell lines. Int J Radiat Biol. 2015, 91(1):81-9.
- It might be better to put the Table with irradiation settings in the main body of the text but not in the Supplementary Material. The authors should explain using plain Gy instead of Gy-Eq for neutron irradiation during BNCT.
- The cytotoxicity study should also show the safe concentration range in the linear graph for the selected compounds to know most preferable still safe concentrations for further biological experiments. Even if the Table 1 is informative, it might be difficult to say, what is the highest though still safe concentration of the compound 19, for example. Please move the graphs from Supplementary Materials to the main file of the manuscript.
- Please justify the use of boron concentration in microgram per mg of protein rather than per number of cells in boron accumulation study.
- Could the authors explain or discuss on the peak of boron concentration at 6 hours after HT-26 cells incubation with Hybrid 19?
- To compare with previously published works on BNCT, the results of irradiation experiments should be presented as fluence-dependent (irradiation dose-dependent) exponential curves, and it would be better to calculate alpha and beta parameters of the linear quadratic equation (LQ-model fit) to show the actual nature of the response to irradiation. The following references will help:
- Sato, E.; Zaboronok, A.; Yamamoto, T.; Nakai, K.; Taskaev, S.; Volkova, O.; Mechetina, L.; Taranin, A.; Kanygin, V.; Isobe, T.; Mathis, B. J.; & Matsumura, A. Radiobiological response of U251MG, CHO-K1 and V79 cell lines to accelerator-based boron neutron capture therapy. J Radiat Res 2018, 59, 101-107. https://doi.org/10.1093/jrr/rrx071
- Yamamoto T, Matsumura A, Yamamoto K et al. Characterization of neutron beams for boron neutron capture therapy: in-air radiobiological dosimetry. Radiat Res 2003;160:70-76. https://doi.org/10.1667/rr3012
- Matsuya, Y.; Fukunaga, H.; Omura, M.; Date, H. A Model for Estimating Dose-Rate Effects on Cell-Killing of Human Melanoma after Boron Neutron Capture Therapy. Cells 2020, 9, 1117. https://doi.org/10.3390/cells9051117
- Franken, N., Rodermond, H., Stap, J. et al. Clonogenic assay of cells in vitro. Nat Protoc 1, 2315–2319 (2006). https://doi.org/10.1038/nprot.2006.339
Otherwise justify why the authors used a simple histogram with % of cell survival, when exponential decrease in surviving fraction is generally accepted as an index of radiotherapy efficacy.
- As synthesized compounds possess a tumor-killing effect additional to one related to neutron irradiation, but BPA only acts during BNCT due to boron neutron capture, this issue should be verified and discussed, and the options to figure out actual boron effect of the synthesized compounds should be provided.
- The article should be proofread including a native English speaker revision to avoid language mistakes/misprints and the use internationally understood vocabulary.
Page 2 Line 64 “It will not only gives…”
Page 2 Line 69 “…which exploite….”
Page 2 Line 73 a/the? “boron cluster…”
Page 2 Line 91 “…reported,.”
Page 7 Line 291 “unpaired T Test were…
Page 7 Line 297 “by seeding glial cells … were….”, etc.
The use of the words “milieu” (instead of “medium”), “promissory” (probably instead of “promising”), “exploite”, etc. should be discussed with a native English editor.
Author Response
Reviewer 2:
Referee: If the authors put discussions in the Results section, they should change the section title to Results and Discussion, if it is allowed by the journal. Otherwise, the article should have a proper Discussion with the comparison of authors’ results with other related studies and listing the limitations, as a general rule.
Answer: The section 3 has been titled as Results and Discussion according to reviewer’s recommendation.
R: The use of the mentioned cell lines should be justified. In different experiments the authors used different cell lines. It is confusing and difficult to make any conclusions based on different experiments with different cell lines in each experiment. For example, U87 MG cells were used in flow cytometry analysis and fluorescence microscopy, whereas F98 or HT-29 cells were used in irradiation experiments. Why did the authors choose those lines, but not others? For example, “to address the issue of ….etc. we used the following cell lines….” etc. Or “these cells are planned to be used in further in vivo experiments”, etc.
A: Following the reviewer’s recommendation, changes are made in the revised manuscript file as detailed below.
On page 10, section 3.2.1, of this revised article file, it is written: “To address the issue of human therapy in vitro cytotoxicities of the hybrids 18-22 and the bioisoster 23 were determined on TKRs-overexpressing cells [18]: Homo sapiens colorectal adenocarcinoma HT-29 and brain glioblastoma U87 MG, and for further animal-in vivo experiments Rattus norvegicus brain glioma C6 were also included in this study (Table 1)”.
On page 12, section 3.3.1, of this revised article file, it is written: “First, we analyzed the presence of boron promoted by 19 and 22 into two different cellular systems, i.e. HT-29 and Rattus norvegicus brain glioblastoma F98 cells to address further in vivo-animal BNCT studies”.
R: Proper information on the equipment/reagents origins should be provided, including cities and countries of the companies (as a general rule).
A: The information has been added at the section 2. Materials and Methods, according to reviewer recommendation.
R: -The ethic committee approval or following the corresponding rules/protocols/conventions should be mentioned when using laboratory animals (here – primary cell culture preparation).
For example: “All manipulations with animals were carried out in compliance with research practices in accordance with the directives of the European Community (86/609 / EEC).”
A: Thanks to the referee for his/her comment that it was really missed in the previous manuscript submission. We have added it on page 6, lines 237-243 of this revised manuscript file.
R:- The type of the primary cell culture should be mentioned in the paragraph title (Primary glial cell culture, for example), as it is not clear from the title and the preparation protocol should be listed if there is any published previously. Is the preparation of the primary cells culture unique in your study or are there any previously published protocols that you used ore modified for that? If protocols exist, they should be mentioned and referenced.
A: This information was added on page 2 according to reviewer recommendation.
R: The protocol for the SRB-based cytotoxicity assay should be mentioned and put in references, as there are numerous publications of the described here cytotoxicity method. The chosen cytotoxicity assay should be justified, for example “to avoid direct interference of the compounds with the reagents…, etc.”, as many researchers use proliferation-based MTT, MTS, or other methods to show live cell rates, but here all cells are stained.
As an example, please refer to: Vichai, V.; Kirtikara, K. Sulforhodamine B colorimetric assay for cytotoxicity screening. Nat Protoc. 2006;1(3):1112-6. DOI: 10.1038/nprot.2006.179
A: The protocol was already mentioned at the Materials and Methods section of the previously submitted manuscript file. Following referee’s suggestion, the reference (31) has been added in this revised manuscript file.
R: The protocol for BPA-fructose solution preparation should be cited, otherwise the authors need to prove that they used their own original protocol for BPA solution preparation, different from the generally accepted one, for example, published by Rossini et al. (2015) or others.
Rossini AE, Dagrosa MA, Portu A, Saint Martin G, Thorp S, Casal M, Navarro A, Juvenal GJ, Pisarev MA. Assessment of biological effectiveness of boron neutron capture therapy in primary and metastatic melanoma cell lines. Int J Radiat Biol. 2015, 91(1):81-9.
A: The suggested reference is incorporated to this revised manuscript; it is reference 35.
R: It might be better to put the Table with irradiation settings in the main body of the text but not in the Supplementary Material. The authors should explain using plain Gy instead of Gy-Eq for neutron irradiation during BNCT.
A: The table with the dosimetry studies has been moved from the S.I. to the main body of the manuscript as suggested by the reviewer.
We use the total physical absorbed dose (in Gy), which is the sum of the specific and nonspecific doses (boron, 14N and gamma doses) produced in the cell line during the application of BNCT. To calculate the Gy equivalent dose we should multiply each component by the corresponding RBE o CBE factors (relative biological effectiveness or compound biological effectiveness). The CBE factor depends on the cell line and the boron compound used. In these studies, we are evaluating and comparing the irradiation with the neutron beam alone or with the neutron beam plus each boron compound (BPA or hybrid 19 or hybrid 22) at two different physical absorbed doses (1 and 2 Gy). Next studies will be addressed to calculate the CBE of the compounds 19 and 22.
R: The cytotoxicity study should also show the safe concentration range in the linear graph for the selected compounds to know most preferable still safe concentrations for further biological experiments. Even if the Table 1 is informative, it might be difficult to say, what is the highest though still safe concentration of the compound 19, for example. Please move the graphs from Supplementary Materials to the main file of the manuscript.
A: We have included the values of IC1, IC20 and IC80 for hybrids 19 and 22 as footnote in Table 1 to provide more information on the cytotoxicity studies.
Figures at the S.I. are related to enzymatic inhibition assays (EGFR inhibition) and not to cytotoxicity studies.
R: Please justify the use of boron concentration in microgram per mg of protein rather than per number of cells in boron accumulation study.
A: In our laboratory we have developed a method to measure boron content by ICPAES from cells previously seeded in plates of 24 wells (Dagrosa et al Thyroid. 2002, 12, 7-12, reference 34 in this revised manuscript file). As it was described, it consists on washing twice with PBS 1X after incubating the cells with boron compounds and then to digest the cells with a small volume (0.5 mL) of formic acid (the number of cells in each well is around 104). In this way, the total digestion of the cells is achieved, and an aliquot is taken to measure boron content and another aliquot to measure proteins by Lowry. This method allows us to evaluate the uptake of boron compounds using a small amount of each one of them.
On the other hand, although the total proteins represent around a 20 % of total cell weight, it is not possible to compare µg of boron by grams of proteins weight (ppm) with µg of boron by 104 cells (ppm). However, we can compare the different boron compounds analyzed (19 and 22) between them and with BPA.
Our method is in concordance with other well-established and recognized protocols used in BNCT studies, i.e. E. L. Crossley, F. Issa, A. M. Scarf, M. Kassiou, L. M. Rendina, Synthesis and cellular uptake of boron-rich pyrazolopyrimidines: exploitation of the translocator protein for the efficient delivery of boron into human glioma cells. Chem. Commun., 2011, 47, 12179–12181; D. Alberti, A. Toppino, S. Geninatti Crich, C. Meraldi, C. Prandi, N. Protti, S. Bortolussi, S. Altieri, S. Aime, A. Deagostino, Synthesis of a carborane-containing cholesterol derivative and evaluation as a potential dual agent for MRI/BNCT applications, Org. Biomol. Chem., 2014, 12, 2457-2467.
We have added a sentence on section 3.3.1 related to this comment.
R: Could the authors explain or discuss on the peak of boron concentration at 6 hours after HT-26 cells incubation with Hybrid 19?
A: We are very interested for these behaviours.
Although the mechanism of cellular uptake of these hybrids has not yet been described, the peak of boron at 6 hours could be due to a modulation of its transporter in the membrane, increasing the rate of uptake once incorporated. As an example of this, Chun-ling Dai et al (Cancer Res 2008; 68: 19) showed that lapatinib significantly enhanced the sensitivity to ABCB1 or ABCG2 substrates in cells expressing ATP-binding cassette (ABC) transporters.
We are planning experiments that allow us to obtain answers about these phenomena. In future works we will study possible mechanisms of entry (endocytosis mediated by receptor or cassette (ABC) transporters) and action of these boron compounds.
R: To compare with previously published works on BNCT, the results of irradiation experiments should be presented as fluence-dependent (irradiation dose-dependent) exponential curves, and it would be better to calculate alpha and beta parameters of the linear quadratic equation (LQ-model fit) to show the actual nature of the response to irradiation. The following references will help:
Sato, E.; Zaboronok, A.; Yamamoto, T.; Nakai, K.; Taskaev, S.; Volkova, O.; Mechetina, L.; Taranin, A.; Kanygin, V.; Isobe, T.; Mathis, B. J.; & Matsumura, A. Radiobiological response of U251MG, CHO-K1 and V79 cell lines to accelerator-based boron neutron capture therapy. J Radiat Res 2018, 59, 101-107. https://doi.org/10.1093/jrr/rrx071
Yamamoto T, Matsumura A, Yamamoto K et al. Characterization of neutron beams for boron neutron capture therapy: in-air radiobiological dosimetry. Radiat Res 2003;160:70-76. https://doi.org/10.1667/rr3012
Matsuya, Y.; Fukunaga, H.; Omura, M.; Date, H. A Model for Estimating Dose-Rate Effects on Cell-Killing of Human Melanoma after Boron Neutron Capture Therapy. Cells 2020, 9, 1117. https://doi.org/10.3390/cells9051117
Franken, N., Rodermond, H., Stap, J. et al. Clonogenic assay of cells in vitro. Nat Protoc 1, 2315–2319 (2006). https://doi.org/10.1038/nprot.2006.339
Otherwise justify why the authors used a simple histogram with % of cell survival, when exponential decrease in surviving fraction is generally accepted as an index of radiotherapy efficacy.
A: In other radiobiological studies of BNCT we usually plot the cell survival data using the linear quadratic model (Neutron Capture Therapy. Wolfgang AG Sauerwein, Andrea Wittig, Raymond Moss, Yoshinobu Nakagawa Editors. Principles and Applications. Studies on the Possible Application of BNCT to Thyroid Carcinoma. Pisarev M, Dagrosa MA, Juvenal G. 2002 Springer. ISBN 978-3-642-31334-9). However, in the present studies, we are comparing a biological effect at two single physical absorbed dose 1 and 2 Gy arising from neutron beam alone o BNCT with the different boron compounds (BPA and hybrids 19 and 22). At these doses we showed a significant decrease of survival fraction for BNCT with hybrids 19 and 22 respect to NCT or BNCT with BPA.
In the literature, results of biological effects for different doses of radiation as histograms are shown. Also, the survival fraction to 2 Gy (FS2) is considered the best parameter to describe the low-dose region on the survival curve and to discriminate the intrinsic radiosensitivity between cell lines and between treatments. Furthermore, from a clinical point of view, this parameter is important as it represents the therapeutically relevant area of the in vitro cell survival curves (Fertil B y Malaise EP, 1981; Deschavanne PJ y Fertl B, 1996).
R: As synthesized compounds possess a tumor-killing effect additional to one related to neutron irradiation, but BPA only acts during BNCT due to boron neutron capture, this issue should be verified and discussed, and the options to figure out actual boron effect of the synthesized compounds should be provided.
A: We consider that this issue was verified with all the initial in vitro studies using cellular systems (tumoral and normal cells) determining the corresponding IC50 for each compound and each cellular systems.
Additionally, in the BNCT experiments the cells were treated in identical conditions (10 ppm equivalent of 10B of each compound, time, etc) but without neutron irradiation verifying 100 % of cellular survival in the treatment with hybrids 19 and 22 (please, see Figure 5 (0 Gy) in the original version of the manuscript).
R: The article should be proofread including a native English speaker revision to avoid language mistakes/misprints and the use internationally understood vocabulary.
Page 2 Line 64 “It will not only gives…”
Page 2 Line 69 “…which exploite….”
Page 2 Line 73 a/the? “boron cluster…”
Page 2 Line 91 “…reported,.”
Page 7 Line 291 “unpaired T Test were…
Page 7 Line 297 “by seeding glial cells … were….”, etc.
The use of the words “milieu” (instead of “medium”), “promissory” (probably instead of “promising”), “exploite”, etc. should be discussed with a native English editor.
A: All language mistakes/misprints above has been amended and the English of this revised manuscript file has been fully corrected by a native English speaker.
Round 2
Reviewer 2 Report
1. The manuscript should be proofread by a native English speaker.
Lines 28-29: “receptors’ inhibition”- “receptor inhibition”
Line 29: “chemiotherapy” – “chemotherapy”
Line 35: “compounds that offering” – “compounds that, offering………combinations), may…”
Line 40: “as agent” – “as agents”
Line 97: “at the literature” should be “in the literature”, etc.
2. We don’t use the word “City” in the equipment/reagent origins, for example: “(Prague City, Czech Republic)” is typically “(Prague, Czech Republic)”, and in other details as well.